# Targeting Tyrosine Kinases in Acute Myeloid Leukemia: Why, Who and How?

**DOI:** 10.3390/ijms20143429

**Published:** 2019-07-12

**Authors:** Solène Fernandez, Vanessa Desplat, Arnaud Villacreces, Amélie V. Guitart, Noël Milpied, Arnaud Pigneux, Isabelle Vigon, Jean-Max Pasquet, Pierre-Yves Dumas

**Affiliations:** 1Institut National de la Santé et de la Recherche Médicale, U1035 Bordeaux, France; 2Université de Bordeaux, 33076 Bordeaux, France; 3CHU Bordeaux, Service d’Hématologie Clinique et Thérapie Cellulaire, F-33000 Bordeaux, France

**Keywords:** acute myeloid leukemia, tyrosine kinase, inhibitors, targeted therapy

## Abstract

Acute myeloid leukemia (AML) is a myeloid malignancy carrying a heterogeneous molecular panel of mutations participating in the blockade of differentiation and the increased proliferation of myeloid hematopoietic stem and progenitor cells. The historical “3 + 7” treatment (cytarabine and daunorubicin) is currently challenged by new therapeutic strategies, including drugs depending on the molecular landscape of AML. This panel of mutations makes it possible to combine some of these new treatments with conventional chemotherapy. For example, the FLT3 receptor is overexpressed or mutated in 80% or 30% of AML, respectively. Such anomalies have led to the development of targeted therapies using tyrosine kinase inhibitors (TKIs). In this review, we document the history of TKI targeting, FLT3 and several other tyrosine kinases involved in dysregulated signaling pathways.

## 1. Introduction

Medicine has traveled a long way since the end of the nineteenth century, when blast cells were characterized in the bone marrow of acute myeloid leukemia (AML) patients. In the last 20 years, the treatment of AML has not kept step with knowledge about the molecular abnormalities leading to leukemogenesis, despite intensive academic and industrial research. While the prognostic classification of AML has improved, therapeutic management is still a matter of debate. Currently, targeted therapies including tyrosine kinase inhibitors (TKIs) are regularly used. Others are still in development to achieve better results in this difficult-to-treat disease.

## 2. Kinome and Tyrosine Kinase

Protein kinases play a key role in signal transduction. About 500 human genes encode kinases while more than 1000 are present in the genome of the popular model in plant biology, *Arabidopsis thaliana* [1,2,3]. Among the kinases, tyrosine kinase (TK) phosphorylation was discovered serendipitously [4]. Until now, about 100 TK have been characterized in humans and are distributed between receptor TK (RTK) and cytoplasmic TK [5]. TK activity is required to phosphorylate substrates on tyrosine residues, including autophosphorylation, allowing the recruitment of signaling partners at docking sites. Tyrosine phosphorylation controls the signaling pathways involved in many cellular processes such as cell growth, proliferation, differentiation and metabolism [6]. Deregulation of the expression and/or activity of TK through mutations or other mechanisms leads to a wide range of diseases and cancers. Blocking its enzymatic activity therefore became a quest for pharmaceutical companies [7,8,9,10]. Around 50 kinase inhibitors are currently FDA approved and referenced in http://www.brimr.org/PKI/PKIs.htm, while at least 150 are being investigated in clinical trials [11,12].

## 3. Tyrosine Kinase in Hematopoietic Tissue

Hematopoiesis in adults subsumes all the biological processes that allow hematopoietic stem cells (HSC) to give rise to all blood lineages in the bone marrow and mature blood cell populations. Several TKs play a key role at different steps of hematopoiesis [13]. To date, at least three have been reported to play a role in early hematopoietic stem and progenitor cell (HSPC) growth and differentiation: (i) the macrophage colony-stimulating factor receptor (M-CSFR) [14]; (ii) the stem cell factor receptor (SCFR, KIT) [15]; and (iii) the FMS-like TK 3 (FLT3) [16]. All other hematopoietic growth factors possess receptors without TK enzymatic activity, although most of them recruit cytoplasmic kinases for signaling. These growth factors and cytokines bind to their own receptors, of which many interact with Janus kinases (JAKs) [17,18]. Other cytoplasmic TKs are involved in signaling pathways downstream of these receptors and JAKs. One family is in close vicinity with JAK: the SRC family TK (SFK).

The SFK, which comprises eight members and three kinase-like SRCs, has been shown to be involved in various lineages of hematopoiesis [19,20,21] such as erythroid and megakaryocyte lineages [22,23]. Six SRC TKs are predominantly expressed in blood cells and disruptions of several SRC TKs induce hematologic abnormalities. While the knock-out (KO) of a single *Src* TK gene is not lethal, the KO of the *Src Lyn* gene induces a phenotype of lupus syndrome, suggesting its key role in the control of immune response [24], and HCK single KO associates with extramedullary hematopoiesis. In contrast, double KO can be lethal [25,26]. LYN is also involved downstream of JAK TK and its inhibition prevents CSF1 or G-CSF-induced proliferation [27,28]. LYN also closely interacts with KIT and its molecular inhibition prevents the SCF-induced proliferation of HSPC [29].

Another TK family, the TEC family, including BMX, BTK, ITK, TEC and TXK, is required not only for the development of B and T cells but also for their specific signaling. TEC TKs were identified in hepatocellular carcinoma by screening a cDNA library [30] and they play a major role in immunity [31]. BTK mutations are associated with a decreased affinity towards phosphoinositides and lead to X-linked immunodeficiency in humans and mice. Over 400 BTK mutations have now been described and are spread throughout the gene [32]. Current research is probing the role of BTK and TEC in AML.

SYK and Zap70 TKs are involved in B and T cell signaling, respectively [33,34]. While SYK is present in many tissues, its role in hematopoiesis and hematological disorders has been reported [35]. Mice KO for *Syk* showed B cell deficiency, abnormal hemostasis and embryonic lethality [36,37].

Other TKs such as the FES/FER family are involved in hematopoiesis, but FES does not seem to be a prerequisite for normal myeloid lineage [38]. It may play a redundant function since mice KO for *Fes* showed only slight differences [39]. FES interacts with cytokine receptors, and PU.1 is one of the transcription factors inducing its expression [40].

## 4. Tyrosine Kinase in Acute Myeloid Leukemia

The deregulation of RTK and TK expression and/or activity is a classical hallmark of cancer cells [41]. Deregulation of TK signaling leads to oncogenic signals and has long been identified in malignant hematopoiesis [41,42,43]. The t(9;22), reciprocal translocation giving rise to the Philadelphia chromosome allows expression of a recombinant chimeric TK protein, BCR-ABL1, which constitutively activates many signaling pathways that are usually tightly regulated, leading to chronic myeloid leukemia (CML) [44]. CML is the main TK-dependent cancer because it was the first to be targeted by an FDA-approved therapy, the anti-ABL TKI imatinib.

In AML, the FLT3 receptor, which is usually present on HSPC, is frequently overexpressed and mutated at diagnosis through an internal tandem duplication (25%, ITD) or point mutation in the TK domain (7%–10%, TKD) (Figure 1). Both induce a constitutive signaling associated with the sustained activation of STAT5, resulting in AML proliferation and survival. Correlated to its overexpression, its activation through ligand binding induces signal activation, depicted in Figure 1. An increase in FLT3-ITD ratio at relapse suggests an advantage for the mutated clone and is associated with a poor outcome in AML patients [45]. TKs are also involved in other hematopoietic disorders such as myeloproliferative neoplasms (MPN) [46,47].

AML represents 3% of cancers and 25% of leukemias [48]. It is a heterogeneous hematopoietic malignancy characterized by a specific molecular landscape. Beyond the recurrent chromosomal abnormalities, a large panel of mutations has been identified and generally comprises eight functional categories: (i) signaling pathways that include genes coding for TK proteins (FLT3, c-KIT, (RAS, PTPN11); (ii) transcription factors (RUNX1, CEBP); (iii) spliceosome complex (SRSF2, SF3B1, U2AF1, ZRSR2); (iv) cohesin complex (STAG2, RAD2); (v) epigenetic modifiers with chromatin-modifying genes (ASXL1, ASXL2, EZH2, MLL) and (vi) genes involved in DNA methylation (DNMT3A, TET2, IDH1 and IDH2); (vii) tumor suppressor genes (TP53, PTEN, PHF6); and (viii) NPM1 mutations. NPM1 mutations are isolated because the mechanisms underlying their leukemogenic pathways are multi-faceted (genomic instability, MYC activation, HOX overexpression, ARF relocation, etc.) [49]. It is commonly accepted that, compared to solid tumors, AML has few mutations, suggesting that other factors, including epigenetic ones, contribute to leukemogenesis.

TK mutations in AML involve mainly FLT3 (30%), c-KIT (5%) and JAK2 (2%) and to a lesser extent JAK1, JAK3 and CSF3R. However, TK negative regulators such as PTPN11 (15%) or CBLC and to a lesser extent PTPN14 and PTPRT are also involved. Mutations affecting RAS proteins, which are downstream of multiple RTKs, are beyond the scope of this review. TKs are also frequently deregulated in AML cells, with the overexpression of unmutated proteins able to activate pro-oncogenic signaling. In AML, SYK is a potential target, as its inhibition led to the differentiation of AML cells in vitro and in vivo [50]. SYK is an interesting target since its overexpression promotes AML transformation and resistance to treatment [51]. In addition, its role in lymphoid malignancies led to the development of the TKIs that are currently being tested in clinical trials and transposed to AML [52].

The FLT3 mutation is by far the most frequent TK mutation in AML, leading to the development of a large panel of TKIs. FLT3 encodes a class III RTK that is well expressed in HSPCs and activates the PI3K/AKT and MAPK pathways upon ligand binding. This RTK family also includes KIT, CSFR and PDGFR, comprising an extracellular domain containing five immunoglobulin domains. FLT3-ITD is the most frequent mutation in AML, involving 4 to 400 base-pair insertions [53,54]. FLT3-ITD is also present in 10% of pediatric AML, 1% of myelodysplastic syndrome (MDS) and 3% of acute lymphoblastic leukemia (ALL). Although the FLT3-ITD mutation is a late event in leukemogenesis, it is an important target for the disease [55]. FLT3-ITD carries a poor prognosis in adult AML patients [56].

## 5. Tyrosine Kinase Inhibitors

The chemical classification of kinase inhibitors has been recently reviewed by R. Roskoski Jr. through the structure of the enzyme-bound antagonist complex. Type I inhibitors bind to the active and inactive protein kinase conformation, type II bind to the inactive protein kinase conformation, type III and IV are allosteric, type V are bivalent and type VI bind covalently to their target [11]. Table 1 shows a panel of TKIs used in AML or in clinical trials and Figure 2 shows targeted TKs.

### 5.1. FLT3 Tyrosine Kinase Inhibitors

Type I inhibitors include midostaurin, gilteritinib and crenolanib, and have activity against the FLT3-ITD and -TKD mutations, while type II inhibitors, which include sorafenib and quizartinib, do not have activity against TKD mutations, as the latter favor the active (DFG-in) protein conformation. Pan TKI first-generation inhibitors such as midostaurin and sorafenib have marginal single-agent activity, even if this postulate may be challenged for allogenic transplant maintenance therapy [57]. Conversely, several FLT3 TKIs such as quizartinib, crenolanib and gilteritinib have single-agent activity that leads to complete or near-complete remission, supporting the rationale for the combination of these agents with cytotoxic chemotherapy. Other studies evaluating quizartinib and gilteritinib in association with various chemotherapy regimens are ongoing, while both TKIs have demonstrated an overall survival (OS) benefit as monotherapy in refractory and relapse (R/R) AML. They are therefore the focus of new development strategies in FLT3-mutated AML. The emergence of resistance is expected through various mechanisms, including intrinsic mechanisms such as the activation of bypass signaling pathways and the activation loop or gatekeeper mutations and extrinsic mechanisms, including cell-to-cell interactions and the secretion of various cytoprotective factors [58,59].

Lestaurtinib (CEP-701 Type I) is a first-generation FLT3 TKI, which inhibits JAK2 WT and mutated in MDS cells. Lestaurtinib is an orally available TKI targeting several RTKs and inhibiting constitutively active FLT3 [60]. It has been used firstly in refractory/relapsed (R/R) AML and it is proposed in newly diagnosed AML [61]. Results are a transient decrease of blast in bone marrow. Both at diagnostic and in R/R AML, it induces a decreased blast count, but responses are short-lived.

Sorafenib (BAY43-9006 Type II) is a first-generation FLT3 TKI and a multikinase inhibitor (RAF, PDGFR, VEGFR, c-KIT, FLT3). As a single agent, sorafenib can induce remission in relapsed FLT3-ITD AML by the downregulation of MCL1 and the upregulation of BIM [62,63]. The most interesting results have recently been shown in the post-HSCT (hematopoietic stem cells transplant) setting (SORMAIN study), probably through immune pathways in addition to the classical pathways [64,65].

Midostaurin (PKC412 Type I) is also a first-generation FLT3 TKI, initially developed to target protein kinase C (PKC). As a type I TKI, it inhibits FLT3-ITD and TKD in vitro and in vivo [66]. It was the first TKI against FLT3 to be FDA approved in April 2017 (Figure 2). The randomized phase III RATIFY study evaluated chemotherapy with or without midostaurin for patients with newly diagnosed FLT3-mutated AML and showed an OS benefit in the midostaurin arm [67].

Sunitinib (SU11248 Type I) is a multitargeted TKI (FLT3, PDGFR, VEGFR, c-KIT). It exerts an equal block on FLT3-ITD and TKD [68]. A phase I study showed better responses in FLT3-mutated AML, although of short duration [69]. In a phase I/II trial, it synergized with cytarabine/daunorubicin in FLT3-ITD but not FLT3 WT AML, which are thought to be the clone which relapses [70].

Quizartinib (AC220, Type II) is a second-generation FLT3 TKI with high selectivity for FLT3 (IC50 ≈ 1 nM), although having activity against c-KIT and PDGFR but with a 10-fold IC50 [71,72]. In clinical trials, it was well tolerated and demonstrated efficacy in improving clinical outcomes as a single agent in R/R AML patients (QUANTUM-R study) [73,74,75,76].

Crenolanib (CP868596, Type I) inhibits FLT3-ITD and -TKD [77,78]. In a phase I study, it demonstrated activity in R/R AML patients and led to 39% complete remission [79,80]. Used as a single agent, a recent study reported bypass mechanisms through adding mutations affecting NRAS and IDH2 [81]. Phase III clinical trials are underway in newly diagnosed FLT3-mutated AML versus midostaurin (NCT03258931) and in R/R FLT3-mutated AML in association with chemotherapy versus chemotherapy alone (NCT03250338).

Gilteritinib (ASP2215, Type I) is an FLT3/AXL inhibitor that also has activity against ALK TK. A phase I/II trial in R/R AML demonstrated a good overall response rate in FLT3-ITD and -TKD AML and less in unmutated FLT3 AML [82,83]. Several trials are underway to test gilteritinib as induction/consolidation treatment. A recent study reported resistance against gilteritinib in AML through TK-independent pathways [84]. In clinical trials, it is well tolerated and demonstrated efficacy in improving clinical outcomes as a single agent in R/R AML patients (ADMIRAL study) [85].

FF-10101 is an irreversible TKI through covalent binding to cyst 695 of FLT3 [86,87]. It shows activity against FLT3-mutated AML cells in vitro and is currently being studied in patients with R/R AML in a phase 1/2 study (NCT03194685). Some FLT3-TKIs are currently in preclinical development (e.g., G-749, TTT-3002), others are in early clinical development (e.g., AKN-028) and some have been withdrawn because of their low activity or adverse pharmacokinetic parameters. Some compounds are called “dual” inhibitors, allowing FLT3 inhibition associated with the inhibition of other kinases, such as AMG925 (CDK4/FLT3), SEL24-B489 (PIM/FLT3), CG806 (BTK/FLT3) or TAK-659 (SYK/FLT3). Gilteritinib is usually reported as a “dual” inhibitor (AXL/FLT3).

### 5.2. KIT Tyrosine Kinase Inhibitors

KIT (CD117) is the receptor for the SCF that is expressed on normal HSPC [15]. It is also expressed in 70% of AML and is mutated in 6%–8%, a rate increasing to 30%–46% in the subset of core-binding factor (CBF) AML [88,89,90,91]. This high frequency of mutated KIT in CBF AML suggests that its targeting could be one way to improve these AML outcomes, even if its role in normal hematopoiesis points to potential hematological toxicity and other side effects. Various mutations can affect KIT, the main ones being in exon 8 (extracellular domain 5) and 17 (intracellular TK domain). KIT mutation in exon 8 is a gain of function mutation (D419) inducing dimerization independently of its ligand, while mutations in exon 17 (D816V/N822) are the major mutation in the kinase domain. Their prognostic values are inconsistent [92,93,94]. In combination with chemotherapy, the KIT TKI dasatinib induces P53-dependent AML cell death [95]. In addition, both dasatinib and radotinib induced the death of AML cells by targeting KIT (Figure 2) [96]. SU5416 and SU6668 are multitargeted TKIs that inhibit VEGFR, KIT and FLT3. Their anti-angiogenic properties suggested a broader effect on AML cells expressing a high level of VEGF. While several studies confirmed their potential for inhibition, they seem to have too short a half-life [97,98].

### 5.3. TAM Tyrosine Kinase Inhibitors

The TAM receptor family includes AXL, TYRO3 and MER [99]. AXL is a class X RTK, which was cloned from CML cells [100]. It is activated by homodimerization upon the binding of its major ligand growth arrest-specific 6 (GAS6), while new ligands have been discovered for MER TK [101,102]. The GAS6/AXL pathway contributes to cell growth, survival, invasiveness, chemotaxis, apoptotic body clearance and immunity [103]. AXL is overexpressed in a wide variety of cancers [104]. In AML, high levels of expression of AXL and GAS6 have been related to poor outcomes [105,106] and in CML to resistance to BCR-ABL1 TKI [107,108,109]. Paracrine AXL activation has been shown to induce AML resistance to conventional chemotherapies and to FLT3-targeted therapy [110,111,112,113]. Recently, we reported a specific mechanism in the hematopoietic niche involving STAT5 and hypoxia, which mediates increased AXL expression and activation in AML cells [114]. The involvement of AXL in oncogenic cooperation in a wide range of malignancies made it the “perfect target” [102,115,116,117].

At least 19 AXL TKIs are in phase I/II or preclinical development for cancer therapies [102,118]. However, only bemcentinib (R428, BGB324) is currently being tested in AML. It has been developed by Rigel Pharmaceuticals and used in breast cancer and chronic lymphoid leukemia (CLL) [119,120]. It is now in a clinical trial in AML, but also in NSCLC (non-small-cell lung carcinoma) in combination with erlotinib [116,121,122,123]. Tyro-3 has been detected in several AML cell lines and its signaling is often deregulated [115,124]. ONO-9330547 has been developed as an AXL and MER TKI. This MER/AXL inhibitor has activity against FLT3-ITD AML cells by blocking the cell cycle through CDK/RB/PLK1 inhibition [125,126]. MER TK is also thought to be an interesting target in AML [127,128]. ONO-7475 was shown to have in vivo activity against both unmutated FLT3 and FLT3-ITD cell lines [129,130]. Cabozantinib is a multikinase inhibitor [131] that has been FDA approved in thyroid and renal carcinoma. Used in AML, it blocked FLT3-ITD and FLT3-TKD F691 mutants and was well tolerated, but it requires further investigation [132]. Recently, DS-1205, a new AXL inhibitor, was developed to overcome AXL-mediated resistance to EGFR-TKI in lung cancer. It may soon be transposed and tested in AML [133].

### 5.4. SYK Tyrosine Kinase Inhibitors

SYK TK is well known for its role in immune receptor signaling. Several studies have demonstrated its implication in AML through an increase in expression/phosphorylation, which was correlated to poor outcomes [50,51,134,135]. Despite the development of many SYK TKIs for immunological or lymphoproliferative diseases, their application in AML has been recently studied and reviewed [136].

Fostamatinib (R788) showed interesting STAT5 inhibition in preclinical studies in FLT3-ITD AML [137]. Entospletinib (GS-9973), a specific SYK TKI (IC50 = 7.6 nM), demonstrated significant single-agent activity, but a stronger effect in combination with chemotherapy in a phase I/II trial in AML [138]. Recent results of a phase II trial in t(11q23.3)/MLL AML and ALL patients reported at an AACR (American Association for Cancer Research) meeting showed a strong effect in monotherapy and a high response rate in combination with chemotherapy [139]. TAK-659 is a SYK/FLT3 TKI that is available for oral administration [140]. A phase 1b/2 clinical trial in R/R AML patients is ongoing (NCT02323113) [141]. Results from clinical trials in lymphoma are very encouraging and deserve further research in AML.

### 5.5. SFK Tyrosine Kinase Inhibitors

Several SRC family kinases (SFKs) are involved in AML (LYN, FYN, HCK, LCK and FGR). In normal hematopoiesis, transient expression and various roles of SFK have been reported such as the blockade of differentiation [142,143]. In AML, LYN is one of the predominant overexpressed SFKs and has been previously targeted using PD166285 [144]. However, different levels of expression are detected in chronic and acute leukemia [145]. In addition to overexpression, LYN is involved downstream of FLT3-ITD through direct interaction with FLT3 [146,147,148]. SFKs are reported to be activated downstream of FLT3-ITD but not FLT3-TKD [149,150]. Various SFKs can activate STAT5 [142] and the combination of SRC and FLT3 TKI is additive only in FLT3-ITD AML cells [151]. To date, dasatinib and ponatinib have been used in AML and have shown activity against FLT3-ITD AML [95,152]. Several clinical trials are ongoing using ponatinib in FLT3-ITD AML consolidation (NCT02428543), with or without azacytidine in FLT3-mutated AML (NCT02829840), or for preventing relapse after HSCT (NCT03690115). New TKIs are under development, such as SAR103168, a multikinase inhibitor inhibiting all the SFKs [153], the TL02-59, a dual inhibitor of FES/FLT3 and FGR TKI (TL02-59) [154,155].

### 5.6. MET/RON Tyrosine Kinase Inhibitors

The two RTKs in this family, MET and RON, have been detected in AML cells [156,157]. They belong to the class VIII RTK family and have a specific pattern of expression, which is overexpressed in a wide range of cancers and in various splicing forms. The MET ligand (HGF) exerts autocrine signaling in AML disrupted by crizotinib. However, resistance may occur and can be bypassed only by using combined therapies that involve inhibition of MET and FGFR [156,158]. MET TKI SU11274 blocked MEIS1/HOXA9-induced leukemia in an AML murine model, suggesting its potential [159]. Another TKI, BMS777607, has a wide range of targets, including TAM-R, RON and MET.

### 5.7. TEC Family Tyrosine Kinase Inhibitors

The TEC family is composed of five members (BMX, BTK, ITK, TEC and TXK) [160]. Bruton’s TK (BTK) was cloned in 1993 and its interaction with KIT was reported [161,162], leading to a mechanistic study in KIT-expressing AML. The BTK TKI ibrutinib, which is already used in CLL, showed that it is worthwhile targeting BTK in AML [163] and particularly in FLT3-ITD AML [164]. In addition, ibrutinib inhibits the SDF1/CXCR4-mediated migration of AML cells [165]. FDA approved for lymphoid malignancies, ibrutinib targeted FLT3-ITD AML cells in preclinical models and showed activity against some TKD mutants [166,167].

## 6. Concluding Remarks

For the last forty years, AML treatment has been limited to intensive chemotherapy with cytarabine and anthracycline. Deciphering the molecular landscape of AML has allowed accurate prognostic classification. One of the more mutated genes in AML is the RTK FLT3 with a frequency around 30%. This has given rise to the development of specific inhibitors to block the signaling of this RTK. The targeted therapies, including TKI, have improved outcomes in R/R AML and are currently being studied in combination with intensive chemotherapies or hypomethylating agents. In addition to the development of TKIs in AML, many other kinases or phosphatases can be targeted. For example, PTPN11, the SHP-2 phosphatase is a regulator downstream of many TKs that could be an additional target, like the similar SHP-1 (PTPN6), which is downregulated in CML and FLT3-ITD AML.

Onco-theranostic strategies are making a comeback in AML but will probably not be sufficient to obtain a definitive cure. Future associations with immune-oncologic strategies could be an interesting option to get the best out of these treatments.

## Figures and Tables

**Figure 1 ijms-20-03429-f001:**
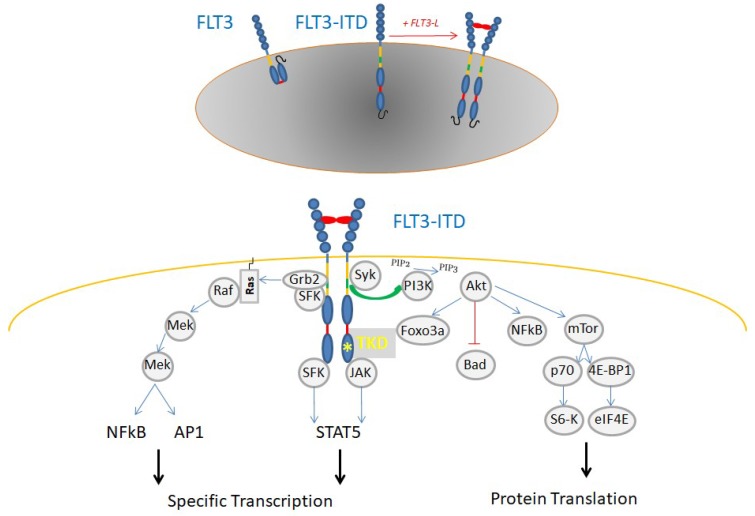
Signaling in AML FLT3-ITD. Like most receptor tyrosine kinases, FLT3 is activated following ligand-induced dimerization and then signals through two major pathways: PI3-kinase/AKT and RAS/RAF/MAP-kinases. Internal tandem duplication by changing structure and localization induces constitutive activation of FLT3 signaling pathway with a large increase in STAT5 activation.

**Figure 2 ijms-20-03429-f002:**
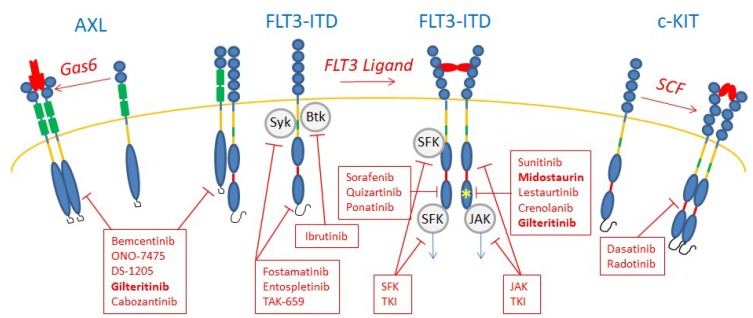
Tyrosine kinase inhibitors in FLT3-ITD AML. Midostaurin and gilteritinib are shown in bold as the only two FDA-approved TKIs. All the others are under development and most are being tested in clinical trials. Those still in preclinical development are not presented.

**Table 1 ijms-20-03429-t001:** TKI-based clinical trials in acute myeloid leukemia (recruiting, active, not recruiting or completed trials).

Drug, TK Targeted and Development Status in AML			NCT Number
Midostaurin	FLT3	FDA approved in newly diagnosed AML	-
Gilteritinib	FLT3	FDA approved in R/R AML	-
Quizartinib	FLT3	MHLW of Japan approved in R/R AML	-
Crenolanib	FLT3	7 phases 1 to 3 studies recruiting or active not recruiting in Clinical trials	-
Sorafenib	FLT3	15 phases 1 to 3 studies recruiting or active not recruiting in Clinical trials	-
Sunitinib	FLT3	Phase 1 and 1/2 recruiting and active	NCT01620216
Lestaurtinib	FLT3	Phase 1/2 completed study in R/R AML	NCT00469859
FF-10101	FLT3	Phase 1/2 recruiting study in R/R AML	NCT03194685
SEL24-B489	FLT3	Phase 1/2 recruiting study in newly diagnosed or R/R AML	NCT03008187
TAK-659	FLT3	Phase 1/2 completed study in newly diagnosed or R/R AML	NCT02323113
Dasatinib	KIT	13 phases 1 to 3 studies recruiting or active not recruiting in Clinical trials	-
SU5416	KIT	Phase 1/2 completed study in newly diagnosed or R/R AML	NCT00005942
Bemcentinib	AXL	Phase 2 recruiting study in newly diagnosed AML	NCT03824080
Cabozantinib	AXL	Phase 1 completed study in newly diagnosed or R/R AML	NCT01961765
Entospletinib	SYK	Phase 1/2 completed study in newly diagnosed or R/R AML	NCT02343939
SAR103168	SFK	Phase 1 completed study in R/R AML	NCT00981240
Ibrutinib	Btk	Phase 1 completed study in R/R AML	NCT02635074

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
