# Peer review of "Targeting Tyrosine Kinases in Acute Myeloid Leukemia: Why, Who and How?"

_ijms, 2019, doi:10.3390/ijms20143429_

Reviewer 1 Report

To give my opinion, I would like to say that it is an excellent review on the importance of targeting FLT3-ITD in the treatment of AML and a up-to-date analysis of the major new drugs in this field.
On the other hand, I suggest some minor changes. I think that the paper should clarify in major details how many AML cases (in percentage) are due to FLT3 changes and what are the molecular mechanisms by which these changes result in the development of AML.
In other words, a  more space should be dedicated to the molecular connection between the genetic change and the development of this hematologic tumor.

Author Response

According to both reviewers, we modified the tittle to " Targeting Tyrosine Kinases in Acute Myeloid Leukemia : Why, who and How? " as this review presents different panel of TK which can be targeted in AML with an obvious part on FLT3.

In agreement to the suggestion of the reviewer, we change the text to emphasize the consequences of FLT3 mutations on signaling and AML proliferation. This has been completed by the percentage of FLT3 mutation occurrence and what the part of ITD and TKD mutations.

an obvious part on FLT3.

 Reviewer 2 Report

This is a well written and clearly stuctured review of the role of tyrosine kinases in hematopoietic malignancies with, as proposed by the title, an emphasis on the role of Flt3 in AML. The review discusses a plethora of inhibitors of tyrosine kinases in AML and is not focused on Flt3, so maybe the title should be more general such as:

Tyrosine Kinase Inhibition in Acute Myeloid Leukemia with emphasis on the specific role of FLT3-ITD Signaling

To be corrected:

lane 213ff ...In AML, levels of expression of AXL and GAS6 have been related to poor outcomes...

I think this sentence should read: In AML, high levels of expression of AXL and GAS6 have been related to poor outcomes...

Author Response

We would thank the reviewer for the comment on the tittle to be more appropriate. Tittle has been modified as

Targeting Tyrosine Kinases in Acute Myeloid Leukemia : why, who and how ?

Best Regards

Reviewer 3 Report

Comments:

#The initial part of the review gives a background of different tyrosine kinases involved in hematopoiesis with  emphasis on Flt3. This is followed by extensive coverage of tyrosine kinase inhibitors. The way the authors approach this text makes it a bit confusing. For example following the section on FLT3 inhibitors the authors discuss other tyrosine kinases and their inhibitors. It is necessary to explain in a small paragraph why other tyrosine kinases and their inhibitors are being discussed and their relevance to FLT3-ITD. For example how are MET/RON inhibitors connected to FLT3-ITD specifically? This has been approached arbitrarily in  subsections, but it needs to be explained earlier and organized better. It is not clear which inhibitors are more or less potent at inhibiting FLT3-ITD specifically. Overall it feels that the text does not do enough justice to the title.

#Some FLT3 inhibitors such as Sunitinib, Lestaurtinib have not been mentioned. Can the authors provide an explanation?

Author Response

We would like to thank the reviewer for his comments.

According to both reviewers, the titlle is not appropriate in regard to the manuscript content and what we would like to present and discuss in this review. The tittle has been changed to Targeting tyrosine kinases in acute myeloid leukemia : why, who and how.

We add a paragraph to explain, even if FLT3 is the major tyrosine kinase targeted, why many other TK can also be interesting targets.

In accordance to the comment why some TKI have not been included in the review such as sunitinib or lestaurtinib, we add in the text why some TKI tested are no longer in development in AML and we  present them in Table 1 and in the text.